# Measured and Estimated Glomerular Filtration Rate to Evaluate Rapid Progression and Changes over Time in Autosomal Polycystic Kidney Disease: Potential Impact on Therapeutic Decision-Making

**DOI:** 10.3390/ijms25095036

**Published:** 2024-05-05

**Authors:** Rosa Miquel-Rodríguez, Beatriz González-Toledo, María-Vanessa Pérez-Gómez, María Ángeles Cobo-Caso, Patricia Delgado-Mallén, Sara Estupiñán, Coriolano Cruz-Perera, Laura Díaz-Martín, Federico González-Rinne, Alejandra González-Delgado, Armando Torres, Flavio Gaspari, Domingo Hernández-Marrero, Alberto Ortiz, Esteban Porrini, Sergio Luis-Lima

**Affiliations:** 1Nephrology Department, Complejo Hospitalario Universitario de Canarias, 38320 La Laguna, Spain; 2Department of Nephrology and Hypertension, IIS-Fundación Jiménez Díaz UAM, 28040 Madrid, Spain; 3Department of Medicine, RICORS2040, 28049 Madrid, Spain; 4Departamento de Medicina, Facultad de Medicina, Universidad Autónoma de Madrid, 28029 Madrid, Spain; 5Laboratory of Renal Function (LFR), Faculty of Medicine, Complejo Hospitalario Universitario de Canarias, University of La Laguna, 38320 La Laguna, Spain; 6Department of Laboratory Medicine, Complejo Hospitalario Universitario de Canarias, 38320 La Laguna, Spain; 7Instituto de Tecnologías Biomédicas (ITB), Faculty of Medicine, University of La Laguna, 38320 La Laguna, Spain

**Keywords:** autosomal dominant, polycystic kidney disease, rapid progression, tolvaptan, glomerular filtration rate decline, measured GFR, estimated GFR

## Abstract

Autosomal polycystic kidney disease (ADPKD) is the most common genetic form of kidney failure, reflecting unmet needs in management. Prescription of the only approved treatment (tolvaptan) is limited to persons with rapidly progressing ADPKD. Rapid progression may be diagnosed by assessing glomerular filtration rate (GFR) decline, usually estimated (eGFR) from equations based on serum creatinine (eGFRcr) or cystatin-C (eGFRcys). We have assessed the concordance between eGFR decline and identification of rapid progression (rapid eGFR loss), and measured GFR (mGFR) declines (rapid mGFR loss) using iohexol clearance in 140 adults with ADPKD with ≥3 mGFR and eGFRcr assessments, of which 97 also had eGFRcys assessments. The agreement between mGFR and eGFR decline was poor: mean concordance correlation coefficients (CCCs) between the method declines were low (0.661, range 0.628 to 0.713), and Bland and Altman limits of agreement between eGFR and mGFR declines were wide. CCC was lower for eGFRcys. From a practical point of view, creatinine-based formulas failed to detect rapid mGFR loss (−3 mL/min/y or faster) in around 37% of the cases. Moreover, formulas falsely indicated around 40% of the cases with moderate or stable decline as rapid progressors. The reliability of formulas in detecting real mGFR decline was lower in the non-rapid-progressors group with respect to that in rapid-progressor patients. The performance of eGFRcys and eGFRcr-cys equations was even worse. In conclusion, eGFR decline may misrepresent mGFR decline in ADPKD in a significant percentage of patients, potentially misclassifying them as progressors or non-progressors and impacting decisions of initiation of tolvaptan therapy.

## 1. Introduction

Autosomal dominant polycystic kidney disease (ADPKD) is characterized by the formation of renal cysts that progressively increase in size and number, replacing normal renal parenchyma, leading to progressive loss of renal function over time, kidney failure and eventual need for kidney replacement therapy (KRT) [1,2]. ADPKD is the most common cause of genetic chronic kidney disease (CKD), mainly caused by genetic variants in one of two genes: PKD1 or PKD2 [3]. Approximately 8% of patients with kidney failure on KRT suffer from ADPKD: the total was around 45,000 persons in Europe in 2021, reflecting the need to improve management and outcomes [4]. There is only one approved treatment for ADPKD: the vasopressin receptor 2 blocker tolvaptan. However, prescription of tolvaptan is limited to persons with rapidly progressing ADPKD [5,6,7], emphasizing the need to estimate rapid progression more accurately, as it may make the difference between being treated or not. In this regard, conventional treatment with blood pressure control and renin–angiotensin system blockers did not appear to decrease or delay the need for KRT [8]. Also, ADPKD patients have been excluded from clinical trials of the SGLT2 inhibitors, novel kidney-protective agents [9,10,11,12].

Several risk factors increase the risk of rapid ADPKD progression, including male gender, early appearance of urologic events, hypertension and PKD1-truncated mutations among the most relevant [13]. In this regard, the rate of progression is heterogeneous between and within patients as the rate of progression may accelerate over time [3,6,14]. Existent definitions for rapid progression in ADPKD usually incorporate age limits and baseline estimated glomerular filtration rate (eGFR), as well as kidney size as surrogates for rapid progression [5,7]. Additionally, they usually incorporate eGFR declines with thresholds to consider rapid loss of GFR ranging from ≥3 to 5 mL/min/1.73 m^2^ per year over 3 to 5 years [5,6,7]. Thus, a reliable evaluation of renal function over time is crucial to properly diagnose these patients’ rapid progression.

Renal function loss is usually estimated by serum creatinine and/or cystatin-C based formulas. More than 70 such equations have been described in the last 70 years, based on creatinine and/or cystatin-C. However, the average error of any formula is about ±30% of measured GFR (mGFR) as recently demonstrated, including for ADPKD patients [15,16]. In this regard, limited information is available on the reliability of formulas to properly evaluate the GFR decline over time in ADPKD patients. Missing a diagnosis of rapid progressors will delay treatment initiation in patients that can benefit from it. By contrast, an incorrect diagnosis of rapid progressors in patients with slow renal function loss, or even stable patients, may expose patients unnecessarily to adverse effects such as liver injury, extreme polyuria and hypernatremia and increase the societal costs of the disease [17].

In the present study, we evaluated the reliability of eGFR in reflecting real renal function decline in a large group of patients with ADPKD with diverse levels of baseline GFR, from advanced CKD to glomerular hyperfiltration, by comparing eGFR decline to mGFR decline.

## 2. Results

### 2.1. Participants

A total of 140 patients of the original group of 234 fulfilled the main inclusion criteria of having at least three mGFR values. The causes of exclusion in the original cohort were diverse: end-stage renal disease with a limited health status; initiation of dialysis or a consecutive transplant; poor health condition, mostly dementia or advanced renal disease with many comorbidities; severe complications that precluded the performance of the test (infectious diseases, cancer, cardiovascular events, etc.); or death. Most patients were female (55%) and mean age was 44.5 ± 14 years (Table 1). Mean baseline mGFR by iohexol clearance was 74 ± 29 mL/min. For the patients enrolled at the HUC, a total of 662 mGFR procedures were determined, 585 validated for progression study (median: 6 years, range: 3–10 y per patient) during a mean follow-up period of 4.0 ± 1.3 y. For those enrolled at FJD, a total of 142 measurements of GFR were performed, with 120 validated for progression study (median: 3, range: 3–4 per patient), over a mean follow-up period of 2.16 ± 0.4 years. The other clinical characteristics are set out in Table 1. For cystatin-C formulas, only 97 cases had at least three serum samples available to measure this marker.

### 2.2. Changes in Renal Function Using mGFR Slopes

A total of 66 out of 140 patients (47%) showed rapid GFR progression (−3 mL/min/y or faster), while 32 cases (22%) showed moderate GFR loss (−3 to + 1 mL/min/y), 24 (17%) showed stable renal function (−1 to +1 mL/min/y) and 18 (13%) improved their GFR (+1 mL/min/y or higher) over time (Figure 1).

#### 2.2.1. eGFR Slopes Assessed by Creatinine-Based Formulas (eGFRcr)

Considering an average of creatinine-based formulas, around 68 out 140 patients (48%) were rapid progressors (−3 mL/min/y or faster), 38 (27%) had moderate eGFR loss (−3 to −1 mL/min/y), 20 (14%) had stable renal function (−1 to +1 mL/min/y) and 15 (10%) improved their eGFR (+1 mL/min/y or higher) over time (Appendix A).

#### 2.2.2. eGFR Decline Assessed by Cystatin-C Based Formulas (eGFRcys)

Considering the average of cystatin-C based equations, around 70 out of 97 patients (72%) showed rapid progression, 10 (11%) moderate eGFR decline, 6 (6%) stable eGFR and 11 (11%) improved eGFR over time (Appendix A).

#### 2.2.3. eGFR Decline Assessed by Creatinine and Cystatin-C Based Formulas (eGFRcr-cys)

Considering the average of cystatin-C based equations, around 72 out of 97 patients (74%) showed rapid progression, 11 (11%) showed moderate GFR decline, 10 (10%) were stable and 6 (6%) improved renal function (Appendix A).

#### 2.2.4. Agreement between mGFR and eGFR Slopes

Creatinine-based formulas: mean CCC for slopes of GFR calculated with creatinine-based formulas was 0.655 for the whole group of patients, ranging from 0.619 (lower 95% confidence interval [CI]: 0.525) to 0.705 (lower CI: 0.628) for the aMDRD and LMrev equations, respectively (Table 2), which indicates poor agreement between eGFR and mGFR slopes. For CKD-EPIcr 2009, CCC was 0.674 and lower CI 0.594. When CCC was determined separately in rapid and non-rapid progressors, mean CCC was 0.739 for rapid and 0.161 for non-rapid progressors, indicating moderate and extremely poor agreement, respectively.

Cystatin-C-based formulas: mean CCC was 0.175, ranging from 0.037 (lower CI: 0.014) to 0.302 (lower CI: −0.181) for the FAScys and EFKC-cys, respectively (Table 2), showing extremely poor agreement between eGFR and mGFR slopes.

Creatinine and cystatin-C-based formulas: mean CCC was 0.321, which indicates very low agreement between eGFR and mGFR slopes.

Bland–Altman LAs between eGFR and mGFR slopes were extremely wide. For example, they ranged from −9.2 to 10.1 mL/min/y for aMDRD; −8.6 to 10.2 mL/min/y for CKD-EPIcr; or −9.1 to 11.6 mL/min/y for MCQ. Similar results were observed for other creatinine-based equations, and even higher LAs were observed for cystatin-C-based formulas (Table 2).

### 2.3. Identification of Rapid mGFR Progressors

Creatinine-based formulas: on average, all analyzed formulas failed to detect rapid progression in 37% of the cases and classified them as those with moderate progression (21%), whereas 15% were misclassified as having stable or improving GFR over time (Appendix A and Figure 1). Table 3 shows ten cases of individual extreme errors of formulas in detecting real GFR decline covering the four GFR decline categories: rapid and moderate progressors, patients with stable GFR and those who improved GFR over time. As an example, the mGFR decline for case 1 was −16.4 mL/min/y, consistent with extremely rapid progression, whereas eGFR loss ranged from −0.7 to 3.2 mL/min/y, i.e., stable or even improving eGFR decline (Table 3). Specifically, CKD-EPIcr 2009 estimated the eGFR slope as +2.4 mL/min/yr.

Cystatin-c-based formulas: on average, all analyzed formulas failed to detect rapid progression in 12% of the cases and classified them as those with moderate progression (3%) whereas 9% were misclassified as having stable or improving GFR over time (Appendix A).

Creatinine and cystatin-C-based formulas: in average, all analyzed formulas failed to detect rapid progression in 12% of the cases and classified them as those with moderate progression (3%), whereas 9% were misclassified as having stable or improving GFR over time (Appendix A).

### 2.4. Identification of Moderate mGFR Progressors

Creatinine-based formulas: on average, eGFR incorrectly classified about half of the cases (54%) and misclassified them as rapid progressors (34%), stable cases (16%) or those who improved GFR over time (4%) (Appendix A and Figure 1). As an example, case 2 had an mGFR slope of −1.8 mL/min/y (moderate progressor), whereas eGFR loss ranged from −6.5 to −12.8 mL/min/y, which means extremely rapid progression (Table 3). Specifically, CKD-EPIcr 2009 estimated the eGFR slope as −8.4 mL/min/yr.

Cystatin-C-based formulas: on average, all analyzed formulas failed to detect moderate progression in 84% of the cases and classified them as those with rapid progression (72%), whereas 12% were misclassified as having stable or improving GFR over time (Appendix A).

Creatinine and cystatin-C-based formulas: on average, all analyzed formulas failed to detect moderate progression in 85% of the cases and classified them as those with rapid progression (79%), whereas 11% were misclassified as having stable or improving GFR over time (Appendix A).

### 2.5. Identification of Patients with Stable mGFR Slope

Creatinine-based formulas: on average, eGFR incorrectly classified stable patients in 82% of the cases and misclassified them as rapid progressors (48%), moderate progressors (23%) or those who improved GFR over time (11%) (Appendix A and Figure 1.) As an example, the mGFR decline for case 3 was −0.3 mL/min/y, i.e., having stable mGFR, whereas eGFR decline ranged from −3.2 to −4.5 mL/min/y (Table 3). Specifically, CKD-EPIcr 2009 estimated the eGFR decline as −3.4 mL/min/yr.

Cystatin-C-based formulas: on average, all analyzed formulas failed to detect moderate progression in 92% of the cases and classified them as those with rapid progression (70%), whereas 22% were misclassified as moderate progressors or improving GFR over time (Appendix A).

Creatinine and cystatin-C-based formulas: on average, all analyzed formulas failed to detect moderate progression in 80% of the cases and classified them as those with rapid progression (60%), whereas 20% were misclassified as moderate progressors or improving GFR over time (Appendix A).

### 2.6. Identification of Patients Who Improved mGFR Slope

Creatinine-based formulas: on average, formulas incorrectly classified these patients in 62% of the cases and misclassified them as rapid progressors (88%), moderate progressors (17%) or those with stable GFR over time (26%) (Appendix A and Figure 1). For instance, case 4 had an mGFR decline of +2.5 mL/min/y, whereas eGFRcr ranged from −1.1 to −6.4 mL/min/y, meaning moderate or rapid GFR loss (Table 3).

Cystatin-C-based formulas: on average, all analyzed formulas failed to detect moderate progression in 67% of the cases and classified them as those with rapid progression (36%), whereas 21% were misclassified as moderate progressors or in those with stable GFR over time (10%) (Appendix A).

Creatinine and cystatin-C-based formulas: on average, all analyzed formulas failed to detect moderate progression in 81% of the cases and classified them as those with rapid progression (47%), whereas 15% were misclassified as moderate progressors or those with stable GFR over time (18%) (Appendix A).

### 2.7. Examples of Misrepresentation of mGFR Slopes by eGFR Slopes

In addition to cases 1 through 4, already shown above (Table 3), discrepancies were noted between eGFRcr and/or eGFRcys in patients who had a similar decline in mGFR. For example, cases 5 and 6 showed that mGFR declines were −4.3 and −3.9 mL/min/y, respectively, whereas eGFRcr declines were on average −8.2 mL/min/y (Table 3). Cases 7 and 8 had similar mGFR declines (−2.1 and −2.5 mL/min/y, respectively), while eGFRcr declines were +1.8 mL/min/y for case 7 and −4.5 mL/min/y for case 8. On the contrary, eGFRcys showed an average eGFR decline about six times faster than the mGFR slope for case 7. Similar results were observed for other cases (Table 3).

### 2.8. Sensitivity Analysis

(a) For the 34 cases under treatment with tolvaptan, the tested formulas, either those based on creatinine or cystatin-C or both had a comparable error in detecting the patterns of evolution of mGFR slopes. In brief, about 35% of the cases with rapid progression were undetected by formulas, and 20 to 50% of the cases without rapid progression (moderate, stable or improvement) were erroneously considered as rapid progressors.

(b) Identification of possible candidates for tolvaptan treatment according to the ERA WGIKD/ERKNet position statement 2021 with modifications [5]. We selected cases with mGFR >25 mL/min/1.73 m^2^ younger than 55 years with a proven fast mGFR decline according to age (−2.5 or −3 mL/min/y or faster) as possible candidates for treatment. In 58 candidates, the results of the tested formulas, using creatinine or cystatin-C or both, failed in detecting rapid progression in around 35% of the cases.

## 3. Discussion

The main finding of our study was that formulas to estimate GFR failed to detect the changes and evolution of renal function over time in patients with ADPKD. This was observed both for formulas using creatinine and/or cystatin-C. In brief: around four out of ten rapid mGFR progressors were not detected by any equation and three out of ten cases were erroneously classified as rapid progressors. Of note, in patients with similar mGFR slopes, the same formulas provided opposite results for eGFR slopes, indicating that the error of formulas may happen at random, making the estimation of renal function unpredictable. Altogether, the results indicate that the utility of eGFR decline in reflecting real GFR decline is very limited and this may influence the prescription of tolvaptan in two ways: underprescription for patients who may benefit from treatment and overprescription in patients who may not derive benefit from it.

We evaluated a group of ADPKD patients who underwent repeated measurements of mGFR by the plasma clearance of iohexol over time. The present study is a continuation of a previous cross-sectional analysis [16] focused on the error of formulas in reflecting GFR. In this study, we analyzed whether that error could also affect the evaluation of GFR changes on follow-up. We tested the most commonly used equations to calculate eGFR and old formulas to evaluate possible improvements in GFR estimations in recent decades. For example, we evaluated the performance of the first formula based on creatinine published in 1957 by Effersoe et al. [18] and the first published with cystatin-C by Le Bricon et al. in 2000 [19]. To avoid artifacts resulting from adjustment by BSA, both eGFR and mGFR were analyzed unadjusted. Finally, to provide a clearer idea of GFR changes over time, patients were classified into four groups: (i) rapid GFR loss; (ii) moderate GFR loss; (iii) stable GFR/minor changes; or (iv) improvement in GFR over time, based on predefined cutoff points based on current guidelines with modifications [5,6,7]. This stratification is necessary to evaluate correctly GFR changes, considering different possible scenarios (stability, improvement, and loss of GFR). Using this stratification, we avoided the error of analyzing the whole group using mean values, in which numerically similar positive or negative slopes will cancel each other and decrease the apparent magnitude of the error. As an example, the mean mGFR decline for the whole group was 1.09 ± 4.7 mL/min/y, which does not represent the changes in renal function in the group.

The most important finding of the study was that eGFR formulas do not properly reflect mGFR changes over time in ADPKD, a population at high risk for CKD progression. We foresee two major aspects of this error: (a) not diagnosing rapid progressors as such and (b) diagnosing rapid progressors as those with stable GFR or with moderate evolution of the disease. Both errors have consequences in day-to-day clinics. On the one hand, formulas failed in the diagnosis of rapid progressors towards advanced CKD. In fact, one in three patients experiencing rapid mGFR decline were not detected by any eGFR formula. This is clearly masking progression of the disease in many patients. Rapid GFR decline over time is a useful clinical marker of the loss of renal mass, together with the increase in total kidney volume. Thus, masking the real progression of GFR loss over time may delay the start of tolvaptan, a therapy that may slow disease progression [20,21,22,23,24]. On the other hand, the formulas considered an un-neglectable group (one-third) of patients with mild, stable evolution, or even those with mGFR improvement over time, to be rapid progressors. This was shown in the analysis of non-rapid progressors, where the reliability of formulas in detecting real mGFR decline was especially limited (very low CCC) with respect to that in rapid-progressor patients. The latter is intriguing, and may indicate the known overestimation of creatinine-based formulas. The individualization of rapid progression may promote the starting of therapies unnecessary in this stage of the disease, also increasing the cost to the health care system.

Another aspect to consider of our work is that the error of formulas was particularly relevant to cystatin-C based equations. We evaluated three families of formulas, CKD-EPI, FAS and EFKC, that use creatinine, cystatin-C or the combination of both markers in their algorithms. Compared with creatinine-based formulas, the errors of the CKD-EPI, FAS and EFKC equations based on cystatin-C were larger. This may indicate that in patients with ADPKD, the error of cystatin-C in reflecting renal function loss over time is even more unreliable than those based on creatinine. The causes of this phenomenon are intriguing. High levels of cystatin-C are associated with aspects different from renal function, like subclinical inflammation [25,26]. On note, renal inflammation in ADPKD has been associated with the expression and accumulation of proinflammatory cytokines, including TNFα, IL-6, IL-8 and MCP1 [27,28,29,30]. Moreover, inflammation and, specifically, the inflammatory cytokine TWEAK, can play a key role in defining the rate of cyst initiation and development in ADPKD [31]. So, it could be speculated that patients with ADPKD have a higher inflammatory status, increasing cystatin-C levels independently of GFR and so underestimating real renal function. This may explain the lower reliability of cystatin-C-based formulas in estimating the loss of renal function over time. In any case, the use of cystatin-C must be considered with caution in this population. In this regard, CST3, the gene encoding cystatin-C, is overexpressed in distal tubular cells in human ADPKD [32,33].

Our study has limitations. A subgroup of patients only underwent the plasma clearance of iohexol on three occasions, particularly those in whom we determined cystatin-C. Three GFR values is the minimum required to calculate slopes. A higher number of data points would yield more precise and accurate slope estimates. However, the results we consistent across eGFR equations and were robust enough to detect the error of eGFR slope in ADPKD patients. Furthermore, they represented real-world clinical practice at two different sites. Also, most of the cases were Caucasians and the results may not be extrapolated to other racial groups.

## 4. Materials and Methods

### 4.1. Patients and Design

We previously published a cross-sectional analysis on the lack of agreement between eGFR and mGFR in a cohort of 226 consecutive ADPKD patients with native kidneys [16]. In the present longitudinal study, we analyzed the reliability of eGFR in reflecting real renal function changes over time. We included all 140 patients that had at least 3 mGFR values, the minimum number of measurements required to properly calculate the decline of GFR. Simultaneously to mGFR, serum creatinine and cystatin-C were determined to calculate eGFR decline over time.

Additional inclusion criteria included age >18 y and clinical stability; that is, absence of acute kidney injury, nephrotoxicity, active infection, cancer or cardiovascular disease during the 3 months before inclusion. Exclusion criteria included allergy to iodine; active malignancy; kidney failure (GFR <15 mL/min); uremia or imminent dialysis; severe psychiatric disease; pregnancy; and lactation. The reasons for patients from the cross-sectional cohort lacking follow-up included kidney failure, kidney replacement therapy, severe comorbidities, clinical instability, loss of follow-up or death.

The study was approved by the clinical ethics committee of Complejo Hospitalario Universitario de Canarias (code CHUC_2018_13) and IIS-FJD (code PIC085-19) and participants signed their informed consent.

### 4.2. Measured GFR

GFR was measured by the plasma clearance of iohexol using either plasma analysis or dried blood spot testing (DBS) as previously explained [34]. On the morning of the study, 5 mL of iohexol (Omnipaque 300, GE-Healthcare, Chicago, IL, USA) was administered intravenously. Then, venous blood samples or finger prick capillary blood specimens were obtained at 120, 180, 240, 300, 360, 420 and 480 min for patients with eGFR <30 mL/min; at 120, 180, 240, 300 and 360 min when eGFR was 30–60 mL/min; or at 120, 150, 180, 210 and 240 min for those with eGFR > 60 mL/min.

### 4.3. Iohexol Determination

Iohexol levels were determined in plasma or DBS by high-performance liquid chromatography (HPLC) at the Laboratory of Renal Function of the University of La Laguna [35]. Procedures performed before September 2017 were measured in plasma, and thereafter in DBS. Both methods are interchangeable, as previously reported [34].

### 4.4. Measured GFR Calculation

The plasma clearance of iohexol was calculated according to a 1-compartment model (CL1) by the formula CL1 = Dose/AUC, where AUC is the area under the plasma concentration time curve from time equalling 0 to infinity. The plasma clearances were then corrected using the Bröchner–Mortensen equation: (0.990778 × CL1) − (0.001218 × CL12) [36].

### 4.5. Estimated GFR

We estimated GFR by using 16 representative formulas based on serum creatinine (eGFRcr: Effersøe, Cockcroft–Gault, aMDRD, MCQ, CKD-EPIcr 2009, LM rev, FAScr, EFKCcr), serum cystatin-C (eGFRcy: Le Bricon, CKD-EPIcy, FAScy, EFKCcy) or both markers (eGFRcr-cy: Ma, Stevens, CKD-EPIcr-cy, FAScr-cy). Adjusting GFR for body surface area (BSA) is a procedure with no clear scientific background. Moreover, it may lead to artificial reduction or increment of GFR in subjects with overweight and obesity and in lean subjects, as recently observed [37]. Therefore, we worked only with unadjusted GFR and the agreement between eGFR and mGFR declines was evaluated with the formulas unadjusted for BSA. When equations provided adjusted eGFR data, we reversed the adjustment by applying the following formula: GFR unadjusted = GFR adjusted × (BSA/1.73). BSA was calculated by the Du-Bois and Du-Bois formula [38].

### 4.6. Calculation of GFR Decline

The decline of GFR was calculated by the following equation:$$Slope=\frac{\sum \left(t-\bar{t}\right)\left(v-\bar{v}\right)}{\sum {\left(t-\bar{t}\right)}^{2}}$$
in which *v* is the value of mGFR or eGFR, and *v* (with macron) is the mean of those values. The time *t* is calculated in years counting the days between the date of the last and the first GFR assessments for each patient and scaling to the average days per year (d/365.25). *t* (with macron) is the mean time calculated with the same scaling factor.

### 4.7. Biochemistry

Creatinine (mg/dL) was measured by isotopic dilution mass spectrometry–traceable creatinine (enzymatic assay) using the Cobas c711 module (Roche Diagnostics, Basel, Switzerland) at Department of Laboratory Medicine, Complejo Hospitalario Universitario de Canarias (Tenerife, Spain). Cystatin-C levels (mg/L) were measured by immunonephelometry using the BNII System (Siemens Healthcare Diagnostics, Munich, Germany), calibrated with ERM-DA471/IFCC, at the same Department of Laboratory Medicine, Complejo Hospitalario Universitario de Canarias (Tenerife, Spain).

### 4.8. Agreement between eGFR and mGFR Decline

The error of eGFR decline was evaluated in two steps: First, we calculated specific agreement statistics for the eGFR-based decline and mGFR-based decline as the concordance correlation coefficient (CCC) and Bland and Altman limits of agreement (LAs). CCC values range from 0 to 1 and combine meaningful components of accuracy and precision. A CCC > 0.90 reflects optimal concordance between measurements. Bland–Altman plots show the relationship between the difference between target and observed measurements and the mean of both values [39]. The smaller the LAs, the higher the degree of agreement between measurements.

Then, patients were classified based on the magnitude and sign of mGFR decline into 4 groups: (a) rapid progressors: GFR loss faster than −3 mL/min/y; (b) moderate GFR loss: −3 to −1 mL/min/y; (c) stable renal function: −1 to +1 mL/min/y; and (d) improvement in GFR over time: higher than +1 mL/min/y. We considered the number and the percentage of cases in which each equation classified patients in the corresponding group or not.

Finally, to facilitate the understanding and the magnitude of the error, we provided clinical examples of the error of eGFR decline in the four groups of patients considering the degree of renal function changes. Also, we provide an appendix with the dataset of GFR decline.

### 4.9. Statistics

Data are expressed as mean±SD or median (interquartile range) depending on the distribution of the data. Comparisons between groups of GFR decline were performed with the chi square test. Statistical analysis was performed with IBM SPSS Statistics 20 (Chicago, IL, USA). For the Bland and Altman test, we used the MedCalc statistical package, version 15.8. For the agreement analyses, we used the statistical package AGP (Agreement Program) v.1.0 (IGEKO, SP) available at www.ecihucan.es/lfr/apps/?dir=agreement_installer. (accessed on 01 May 2024). The AGP is based on the R code originally developed by Lawrence Lin and YuYue [40]. The AGP was developed to simplify the use of the tool given in the R agreement package.

### 4.10. Sensitivity Analysis

The performance of formulas in reflecting mGFR decline was tested in two subgroups of patients: (a) those receiving Tolvaptan and (b) possible candidates to receive treatment with Tolvaptan according to current guidelines [5]. In a sub-analysis, we separately determined CCC between mGFR and eGFR declines in rapid and non-rapid progressors for creatinine-based formulas.

## 5. Conclusions

In conclusion, eGFR declines may misrepresent mGFR decline in ADPKD in a significant percentage of patients, potentially misclassifying them as progressors or non-progressors and impacting decisions of initiation of specific therapy with tolvaptan. Moreover, the error was unpredictable, since one-third of the cases who progressed rapidly towards CKD were not detected by formulas, and another one-third were erroneously classified as rapid progressors when actually they were not. The study of the impact of this error in clinical care, and importantly in clinical trials where GFR changes are the main outcome, deserves more attention.

## Figures and Tables

**Figure 1 ijms-25-05036-f001:**
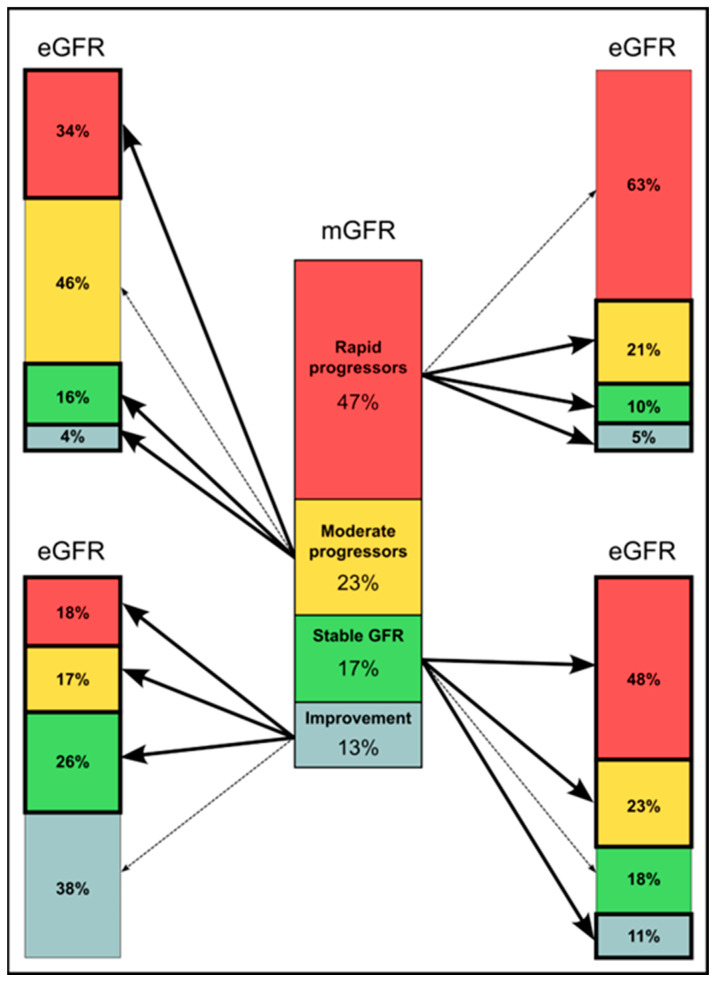
Misclassification of GFR changes over time according with mGFR and eGFR using creatinine-based formulas. Central column: classification of rapid (red) and moderate progressors (yellow), patients with stable GFR (green) or those who improved GFR over time (blue). Columns to the right and left represent the average error of the 8 creatinine-based equations analyzed in detecting each group. In example, the upper right column shows the percentages of real rapid progressors based on mGFR that creatinine-based formulas correctly classified as rapid (red), indicated by a dotted line, or misclassified as moderate progressors (yellow), patients with stable GFR (green) or those who improved GFR over time (blue), indicated by solid lines. The same can be applied to the upper left column (real moderate progressors), lower right column (stable GFR) and lower left column (GFR improvement).

**Table 1 ijms-25-05036-t001:** Baseline clinical characteristics of 140 participants. Data expressed and Mean ± SD or median interquartile range (IQR). ACEI: angiotensin-converting enzyme inhibitor; AR: angiotensin receptor blocker; GFR: glomerular filtration rate; eGFR: estimated glomerular filtration rate; ADPKD: autosomal dominant polycystic kidney disease. * missing data in 7 cases.

N			
	Center 1 (HUC)/Center 2 (FJD), n (%)		102 (73)/38 (27)
	Age (y)		44.5 ± 14
	Women, n (%)		77 (55)
	Weight (kg)		78 ± 16
	Height (cm)		170 ± 9
	Body Mass Index (kg/m^2^)		27 ± 4.6
		Body Mass Index > 30 kg/m^2^, n (%)	30 (21)
**Comorbidities, n (%)**			
	Hypertension		96 (69)
		Diuretics	26 (18)
		Beta blockers	12 (8.5)
		Calcium antagonist	15 (11)
		ACEI	27 (19)
		ARB	62 (44)
		Sympatholytic agents	9 (6)
	Dyslipidemia		35 (25)
		Fibrates	4 (3)
		Statins	26 (18)
	Diabetes		9 (6)
	Hyperuricemia		28 (20)
	Smoking *		
		Never	52 (65)
		Former	27 (19)
		Current	23 (16)
**Renal function**			
	Measured GFR (mL/min)		74 ± 29
		Measured GFR categories, n (%)	
		>90 mL/min (G1)	47 (33)
		60–90 mL/min (G2)	44 (31)
		45–60 mL/min (G3)	21 (15)
		30–45 mL/min (G4)	17 (12)
		<30 mL/min (G5)	11 (8)
	Serum creatinine (mg/dL)		1.25 ± 0.6
	Serum cystatin-c (mg/L)		1.26 ± 0.6
	Proteinuria (mg/24 h)		123 (90–180)
	eGFR MDRD (mL/min)		69 ± 32
	eGFR CKD-EPIcr 2009 (mL/min)		83 ± 33
**ADPKD**			
	Age at diagnosis (years)		33 (22–46)
	Family history, n (%)		115 (82)
	Total kidney volume echo or MRI (mL)		620 (350–1240)
	Patients on tolvaptan, n (%)		34 (24)

**Table 2 ijms-25-05036-t002:** Concordance correlation coefficients (CCC) and Bland and Altman limits of agreement (LAs) between eGFR slopes estimated using different equations and mGFR slopes. CCC data presented as mean (lower 95% confidence interval). The equation most widely used in clinical practice is shown (CKD-EPIcr 2009) in bold font.

	eGFR Equation	CCC	LA (mL/min/y)	n
Creatinine	Effersoe	0.632 (0.540)	−9.2 to 9.1	140
	Cockcroft–Gault	0.631 (0.550)	−9.5 to 11.8	140
	aMDRD	0.619 (0.525)	−9.2 to 10.1	140
	MCQ	0.646 (0.570)	−9.1 to 11.6	140
	**CKD-EPIcr 2009**	**0.674 (0.594)**	**−8.6 to 10.2**	**140**
	LMRev	0.705 (0.628)	−8.4 to 8.6	140
	FAScr	0.637 (0.550)	−9.3 to 10.4	140
	EKFCcr	0.696 (0.619)	−8.5 to 9.3	140
Cystatin-C	LeBricon	0.120 (0.030)	−14.0 to 22.7	97
	CKD-EPIcys	0.243 (0.138)	−10.9 to 21.6	97
	FAScys	0.037 (−0.014)	−15.1 to 25.6	97
	EKFCcys	0.302 (0.181)	−14.0 to 22.7	97
Creatinine and Cystatin-C	Ma	0.248 (0.126)	−9.9 to 18.1	97
	Stevens	0.294 (0.158)	−10.7 to 18.7	97
	CKD-EPIcr-cys	0.347 (0.218)	−8.6 to 15.6	97
	FAScr-cys	0.396 (0.263)	−8.6 to 14.2	97

**Table 3 ijms-25-05036-t003:** Individual examples of measured GFR (mGFR) and estimated GFR (eGFR) declines in ml/min/year. Positive slopes have a blue background of the cell and declines faster than −10 mL/min have a reddish background. The most commonly used equation in the clinic (CKD-EPIcr 2009) is in bold font.

	Case	1	2	3	4	5	6	7	8	9	10
	**mGFR**	**−16.4**	**−1.8**	**−0.3**	**2.5**	**−4.3**	**−3.9**	**−2.1**	**−2.5**	**0.1**	**−0.3**
Creatinine	Effersoe	1.9	−6.5	−3.5	−3.9	−7.4	3.4	3.6	−4.1	−14.2	2.0
	CG	−0.7	−8.6	−3.2	−5.8	−8.1	4.3	2.0	−3.9	−12.6	2.0
	aMDRD	1.7	−7.1	−3.3	−5.2	−8.9	3.2	3.8	−5.9	−15.1	1.9
	MCQ	1.0	−12.8	−4.5	−1.1	−9.9	0.7	−1.4	−0.9	−0.6	0.5
	**CKD-EPIcr 2009**	**2.4**	**−8.4**	**−3.4**	**−6.4**	**−9.3**	**1.3**	**0.0**	**−7.3**	**−5.3**	**2.1**
	LMRev	2.4	−8.0	−3.4	−3.6	−6.2	1.8	1.8	−3.7	−8.2	1.6
	FAScr	2.1	−7.6	−3.9	−4.5	−8.1	3.6	4.1	−4.6	−15.6	2.2
	EKFC_cr	3.2	−8.4	−3.2	−5.0	−7.6	1.0	0.7	−5.4	−4.9	1.7
Cystatin-C	LeBricon	−7.9	−1.5	−3.6	0.9	−7.9	−8.5	−17.3	5.9	−10.8	−4.2
	CKD-EPIcys	−10.3	−2.1	−3.2	−0.1	−4.3	−9.5	−9.9	6.6	−12.6	−6.1
	FAScys	−8.8	−1.8	−4.4	1.0	−9.0	−9.3	−19.5	6.5	−11.7	−5.0
	EKFCcys	−7.3	−2.6	−3.2	0.1	−1.9	−9.4	−8.4	5.6	−9.3	−5.2
Cre and Cys-C	Ma	−6.3	−5.2	−4.1	−2.5	−12.5	−9.2	−11.5	−6.7	−16.8	−2.0
	Stevens	−5.5	−4.9	−3.6	−2.6	−11.0	−8.0	−9.2	−7.2	−14.6	−1.5
	CKD-EPIcr-cys	−5.6	−4.9	−3.5	−2.5	−8.4	−8.7	−7.0	−5.7	−10.8	−2.3
	FAScr-cys	−3.9	−4.3	−4.6	−1.8	−8.6	−8.6	−7.3	−3.6	−13.3	−1.4

## Data Availability

Data can be made available upon request.

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
