# Peer review of "Measured and Estimated Glomerular Filtration Rate to Evaluate Rapid Progression and Changes over Time in Autosomal Polycystic Kidney Disease: Potential Impact on Therapeutic Decision-Making"

_ijms, 2024, doi:10.3390/ijms25095036_

Round 1
Reviewer 1 Report
Comments and Suggestions for Authors
The aim of this manuscript is to compare eGFR versus mGFR in detecting rapid renal function deterioration in patients with ADPKD. The authors involved 160 ADPKD patients and utilized mGFR as the gold standard to determine rapid renal function loss. Various eGFR calculation methods were applied to these patients, revealing that eGFR is not reliable in detecting renal function changes in ADPKD patients, particularly in cases of rapid deterioration.
Several comments are provided below:
- The topic addressed in this manuscript appears to overlap with a study by Rodriguez et al. published in the Journal of Nephrology (2022), which reached a similar conclusion. Are these two studies related? What is the novel finding in this study?
- More detailed figure and table legends should be provided, particularly for tables 2 and 3.
- Further investigation is needed regarding Figure 1 to elucidate the role of eGFR in detecting renal function decline in ADPKD patients. For instance, separating patients into groups based on the degree of renal reduction (rapid, moderate, and mild) and assessing the concordance correlation coefficient (CCC) in each group could provide valuable insights.
Author Response
Please see the attachment the cover letter with answers to reviewers in bold

Reviewer 2 Report
Comments and Suggestions for Authors
I think this is a significant paper, as it warns us that if we evaluate whether or not therapeutic intervention is required for the decline in renal function in ADPKD patients based solely on the decline with estimated GFR (eGFR), we may not be making the right decisions. Additionally, you examined 16 deferent formulas for calculating eGFR and shows that it is difficult to evaluate using any one method, which I think is another important point.
There are some points that are somewhat unclear, and I think the paper would be even better if you could correct them.
Major Revision
Fig1: I think figure 1 is the most important figure, but it is difficult to understand. The middle figure shows the percentage of each category by measured GFR (mGFR), but it would be easier understand if you write down what each the four eGFR figures represents. For example, “The upper right figure shows……” Also, please include an explanation of the practice and the dotted line. The “moderate progressor” and “stable GFR” percentage of mGFR are different between the text and the figure.
Similarly, in “2.2.4 Agreement between mGFR and eGFR slope”, the number in the text and the number in Table 2 do not match.
Table 3: Why did you choose these 10 patients? Please include that explanation as well.
Is this difference between eGFR and mGFR something that only happens in patients with ADPKD? Also, you write about the involvement of cytokines and inflammation with regard to Cyst C. Is there any possible cause for the discrepancy with regard to Cr?
Since eGFR is calculated from Cr, is there a risk that the difference in eGFR will be included in the measurement error of Cr? In other words, the change in eGFR of about 1 to 3mL/min is very small change in Cr value. Perhaps the movement of eGFR 1mL/min will probably be about 0.03-0.05mg/dL.
Your paper is that eGFR cannot be used to determine the rate of decline of renal dysfunction in ADPKD, is there a better other way? Is it only mGFR? If you have any ideas, please write them down!
Minor Revision
Key Words: Is “estimated GFR” not necessary?
Line 94: (median: 6 year, range 3-10 years per patient) → (median: 6, range 3-10 years per patient)?
Line 101: Mean SD → mean±SD?
Line 102: IQR → interquartile range (IQR)
Page 4: Why does show values “eGFR MDRD” and “eGFR CKD-EPIcr 2009” but not value for cyst C eGFR formula in Table1?
Page5 Table 1: “Total kidney value eco (ml)” What is “eco”? “echo”?
Author Response

(The authors gave the same response as above.)

Reviewer 3 Report
Comments and Suggestions for Authors
Thank you for the possibility to evaluate the manuscript entitled: Measured and estimated GFR to evaluate rapid progression and GFR changes over time in ADPKD: potential impact on therapeutic decision-making.
The main question addressed by the research is how to evaluate the reliability of eGFR in reflecting real renal function decline in a large group of patients with ADPKD with diverse levels of baseline GFR, by comparing eGFR decline to mGFR decline.
The main finding of the study was that formulas to estimate GFR failed to detect the changes and progression of renal function impairment over time in patients with ADPKD. This was observed both for formulas using creatinine and /or cystatin-c, which is very informative as compared with other published material. Authors confirmed “Compared with creatinine-based formulas, the error of CKD-EPI, FAS and EFKC equations based on cystatin-C was larger” – this statement was quite a big surprise for me.
Authors added to the subject area the results of examination in 140 patients with different stages of CKD, different comorbidities and varied antihypertensive medications with repeated measurements. The limitations of the study were presented clearly.
The conclusions are consistent with the evidence and arguments presented.
It is very important to have such a report in library to make the diagnosis of rapid deterioration more reasonable and to start the treatment with tolvaptan on right time.
The references are appropriate and in sufficient number. Tables and are carefully prepared.
The manuscript is written very well. I have no further remarks.
Author Response

(The authors gave the same response as above.)

Round 2
Reviewer 1 Report
Comments and Suggestions for Authors
The authors response all the reviewer's comments with modification of the manuscript. The manuscript should be considered to published.
Reviewer 2 Report
Comments and Suggestions for Authors
Thank you for understanding my opinion and for changing your paper. Your paper was clear and easy to understand.
Thank you for your hard work.